# Controls of Seasonal and Interannual Variations on Soil Respiration in a Meadow Steppe in Eastern Inner Mongolia

Xu Wang [1,*], Kaikai Fan [1], Yuchun Yan [1], Baorui Chen [1], Ruirui Yan [1], Xiaoping Xin [1,*] and Linghao Li [2]

1    Institute of Agricultural Resources and Regional Planning, Chinese Agricultural Academy of Sciences, Beijing 100081, China

2    Institute of Botany, The Chinese Academy of Sciences, Beijing 100093, China

\*    Correspondence: wangxu01@caas.cn (X.W.); xinxiaoping@caas.cn (X.X.); Tel.: +86-010-8210-9618 (X.W.); +86-010-8210-9615 (X.X.)

**Abstract:** Understanding long-term seasonal and interannual patterns of soil respiration with their controls is essential for accurately quantifying carbon fluxes at a regional scale. During the period from 2009 to 2014, an automatic measurement system (LI-8150, Licor Ldt., Lincoln, NE, USA) was employed for the measurement of soil respiration in a meadow steppe of eastern Inner Mongolia. We found that the seasonal pattern of soil respiration was controlled mainly by the soil temperature, which explained about 82.19% of the variance. Annual soil respiration varied between 391.4 g cm$^{-2}$ and 597.7 g cm$^{-2}$, and significantly correlated with soil moisture, suggesting that soil moisture was the most predominant factor controlling the annual variations of soil respiration in this meadow steppe. A double factorial exponential model including both soil temperature ($T_S$) and soil water content (SWC) (y = 6.084 × exp(0.098 $T_S$ × SWC) − 5.636) explains 72.2% of the overall variance in soil respiration. We also detected a temporal inconsistency of 2–3 months in the effects of precipitation on soil respiration versus canopy biomass production, which was presumably a main mechanism explaining the weak relationships between soil respiration and phytomass components in this ecosystem. Our findings have important implications for better understanding and accurately assessing the carbon cycling characteristics of terrestrial ecosystems in response to climate change in a temporal perspective.

**Keywords:** soil respiration; grassland; interannual; precipitation; antecedent effects

## 1. Introduction

Soil respiration, an efflux of microbe- and plant-emitted carbon dioxide from soil to the atmosphere, is the second-largest carbon efflux in the global carbon cycle, next only to that from the sea [1]. Therefore, any of its minor changes would pose substantial impacts on the carbon cycling on Earth [2]. Grasslands are one of the main types of territorial ecosystems, occupying 40.5% of the land area, and storing 34% of the terrestrial carbon stock [3]. Given its broad distribution range, bulky soil carbon stock, and sensitivity to climate change and anthropogenic action, the spatio-temporal variations of the global temperate grassland ecosystem as a whole should be rather substantial and multi-factor controlled [4]. Currently, the dynamic pattern of soil respiration as well as its controlling mechanisms remain to be well understood both at varying temporal and spatial scales worldwide [5–7]. Of these, the temporal trends across years or decades based on the observational annual mean values of climate, soil, and biotic drivers with respect to soil respiration are the most poorly documented, which produces a significant problem in more accurately quantifying and predicting carbon fluxes at various spatial scales of the terrestrial ecosphere [8].

Soil respiration is comprised of both autotrophic (plant roots) and heterotrophic (microbes and soil fauna) respiration fluxes, and it is thus an integrative indicator of plant production and rhizosphere organism activity [8]. The seasonal pattern of soil

respiration is predominantly related either to temperature or soil moisture/precipitation, depending on the vegetation type, edaphic traits, and site-specific climate conditions [9–12]. The vegetation phenology and variability of soil moisture also play important roles in controlling the seasonal trajectory of soil respiration [13–15]. In the interannual scale, previous studies showed that soil respiration variation is more substantially dominated by soil moisture or is rainfall-dependent [14,16,17]. Recent reports highlighted that the long-term interannual variability of soil respiration cannot be fully explained by climatic factors alone, but eco-physiological factors such as GPP, phenomenon, and fine root turnover should also be taken into account [18–21]. All this evidence indicated that the patterns of soil respiration may be dominated by different driving factors at different timescales. Therefore, long-term observation is still required to improve the more accurate estimation of spatially and temporally upscaled fluxes [22].

Grasslands occupy some 41% of the total land area in China and have strong spatial heterogeneity in vegetal, edaphic, and climatic conditions. Of these, the meadow steppe is the highest in net primary production, plant diversity, and soil carbon stock [15]. In the country, studies on soil carbon fluxes focus largely on the typical steppe, whereas those concerning the meadow steppe are much fewer. Moreover, most of these studies deal with the seasonal patterns and controls of soil respiration, whereas a few have examined the interannual patterns and the relevant underpinning mechanisms of soil carbon fluxes. In this study, consecutive field observations across six years were conducted in a meadow steppe ecosystem of eastern Inner Mongolia, the objectives of which were as follows: (1) to examine the seasonal patterns and controls of soil respiration in this intact ecosystem; (2) to explore the interannual dynamics, relations, and controls of soil respiration.

## 2. Materials and Methods

### 2.1. Site Description

The study site is located in Hulunbuir, northeastern of Inner Mongolia, China. The region is dominated by a temperate semi-arid continental climate characterized by a lengthy, cold winter; short, muted summer; dry, windy spring; and autumn with early frost and a sudden drop in temperature. The mean annual temperature varies between $-5$ and $0$ °C, with great temperature differences between day and night and among seasons. The average temperature of the coldest month (January) is between $-18$ and $-30$ °C, whilst that of the hottest month (July) is between 16 and 21 °C. The accumulated temperature ($\geq 10$ °C) is 1780–1820 °C, with a frost-free period of 85 to 155 days. The precipitation is highly variable both seasonally and interannually, with an annual precipitation amount of 350 to 450 mm, 75% of which falls during the period from June to September. The vegetation is dominated by *Leymus chinensis*, *Stipa baicalensis*, and *Filifolium sibiricum*. The predominant soils are chernozem and dark chestnut, and SOC averages 2.60% [23].

### 2.2. Field Observation

The experiment was carried out in the Hulunbuir Grassland Ecosystem Research Station (N 49°19′, E 120°03′, alt. 628 m). The experimental plot occupies an area of about $3.34 \times 104$ m$^2$, which has been enclosed since the end of 2006. Before being enclosed, the field was grazed by dairy cows seasonally or year-round. Field measurements were conducted from 2009 through 2015. The soil respiration rate was measured continuously almost every day during the six study years, with a daily frequency of once every two hours. An automatic measurement system (Li-8150, LI-COR, Inc., Lincoln, NE, USA) was employed for the measurement, which consisted of an infrared analyzer, a multiplexer, and six portable chambers (20 cm in diameter). The chambers were set randomly in the plot sward and kept at least 10 m from one another. The PVC chamber mounted on a metal frame was inserted into the soil surface at a depth of 3 to 5 cm to ensure its tightness for the air-proof purpose. The vegetation and plant detritus present inside the chamber-covered spots were removed regularly during each growth season, whilst maintaining the soil surface thereof as intact as possible. The environmental factors such as

air temperature, humidity, pressure, soil temperature, soil water content, and rainfall were also monitored simultaneously by a nearby automated meteorological station (AWS310, VAILASA, Vantaa, Finland).

### 2.3. Aboveground Biomass, Litter, and Root Biomass

Aboveground biomass and litter mass were harvested from three quadrats (1.0 m × 1.0 m) near each plot at the end of July every year. Simultaneously, we collected three soil cores in each quadrat (depth in 0~50 cm) using an 8 cm diameter soil auger. Roots were washed with tap water in the laboratory, and oven-dried at 65 °C for 48 h to a constant weight.

### 2.4. Statistical Analysis

The ratio of the standard deviation to the mean (CV) provides a standardized dispersion of soil respiration measurement. Linear or exponential regression analyses were used to pose the relationships between soil respiration and soil temperature, soil moisture, precipitation, aboveground biomass, litter, and belowground biomass. A one-way ANOVA with Duncan's test was used to examine the significant difference of annual soil respirations from 2009 to 2014. All statistical analyses were conducted by the software SPSS11.0 (SPSS Inc., Chicago, IL, USA). The chart was drawn by Origin 8.0 (OriginLab Ltd., Northampton, MA, USA).

## 3. Results

### 3.1. Seasonal Pattern of Soil Respiration

Generally, the air temperature displayed a regular single-peaked seasonal pattern, with the maximum occurring in July, regardless of the year. By contrast, rainfall seasonally changed much more drastically, usually with multiple peaks in the growing season (Figure 1a). In addition, the peak values of the annual air temperature were much less variable among the years, whereas the peak values of rainfall differed substantially both within a growing season and among the years. As a result, soil respiration exhibited a comparatively irregular seasonal pattern, characterized by several peak values in a single growing season (Figure 1b). The CVs for soil respiration generally varied between 0.39 and 0.54 during the growing seasons of different years.

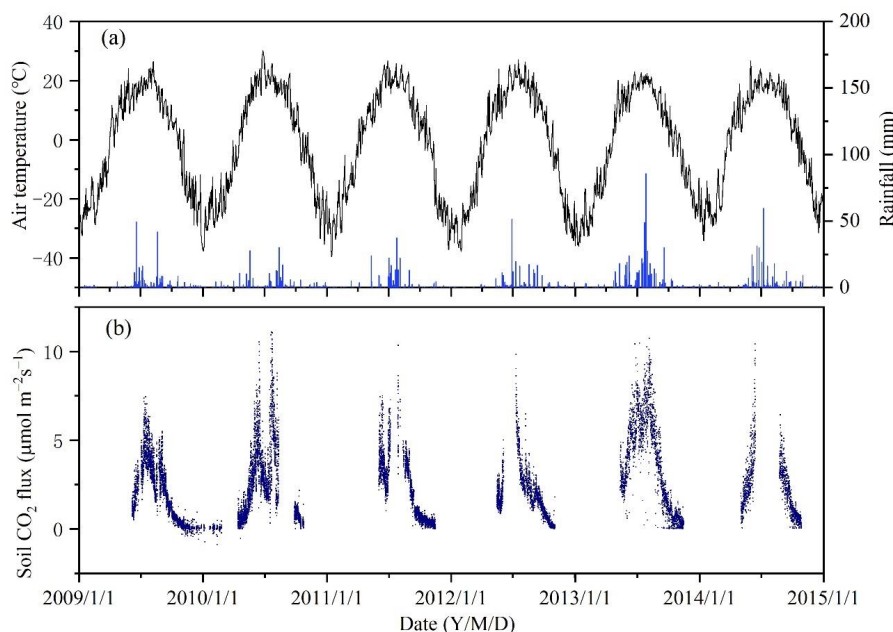

**Figure 1.** Seasonal variations of air temperature, rainfall (**a**), and soil respiration (**b**) from 2009 to 2014 in Hulunbuir meadow steppe.

Our analysis revealed significant positive relationships between soil respiration and soil temperature, both in the growing season of each year and in the entire study period of six years as a whole (Figure 2), which can be best delineated by the exponential function. Meanwhile, positive linear relationships between soil respiration and soil moisture within the years were also marked (Figure 3). However, the determination coefficient values ($R^2$) of soil moisture were much lower than those of soil temperature for both individual years and all years as a whole. When a double factorial exponential model, including both soil temperature and soil moisture, which stands as $Rs = 6.084 \times \exp(0.098 \times Ts \times SWC) - 5.636$ (Rs denotes soil respiration; Ts denotes soil temperature; SWC denotes soil moisture), was employed to modulate the seasonal pattern of soil respiration of all years together, the two climatic factors collectively may explain 72.2% of the overall variance of soil respiration. By a stepwise regression analysis, soil temperature may explain about 82.19% of the overall variance, whilst the remainder 17.81% was explainable by soil moisture.

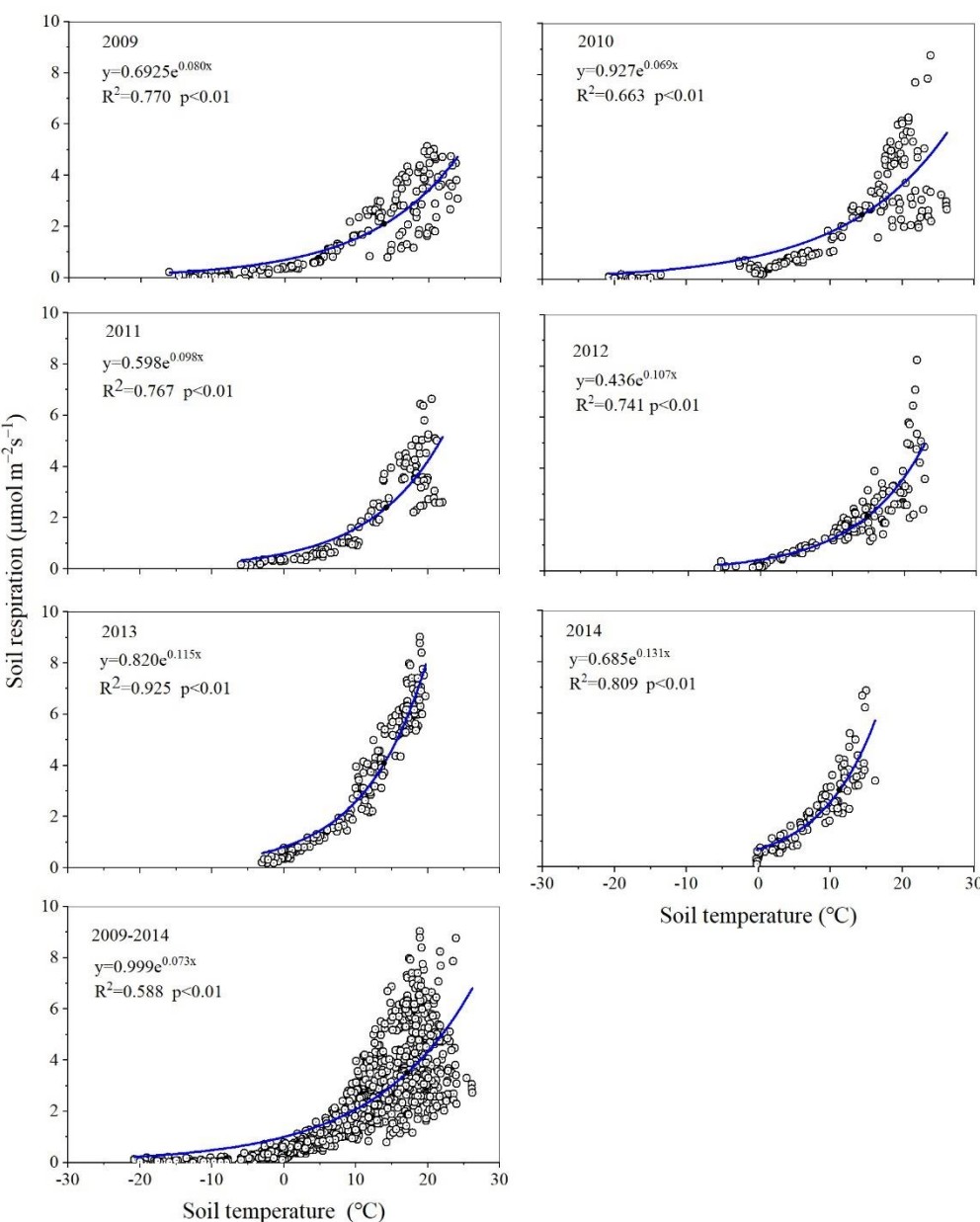

**Figure 2.** Relationship between soil respiration and soil temperature at 10 cm depth during 2009–2014.

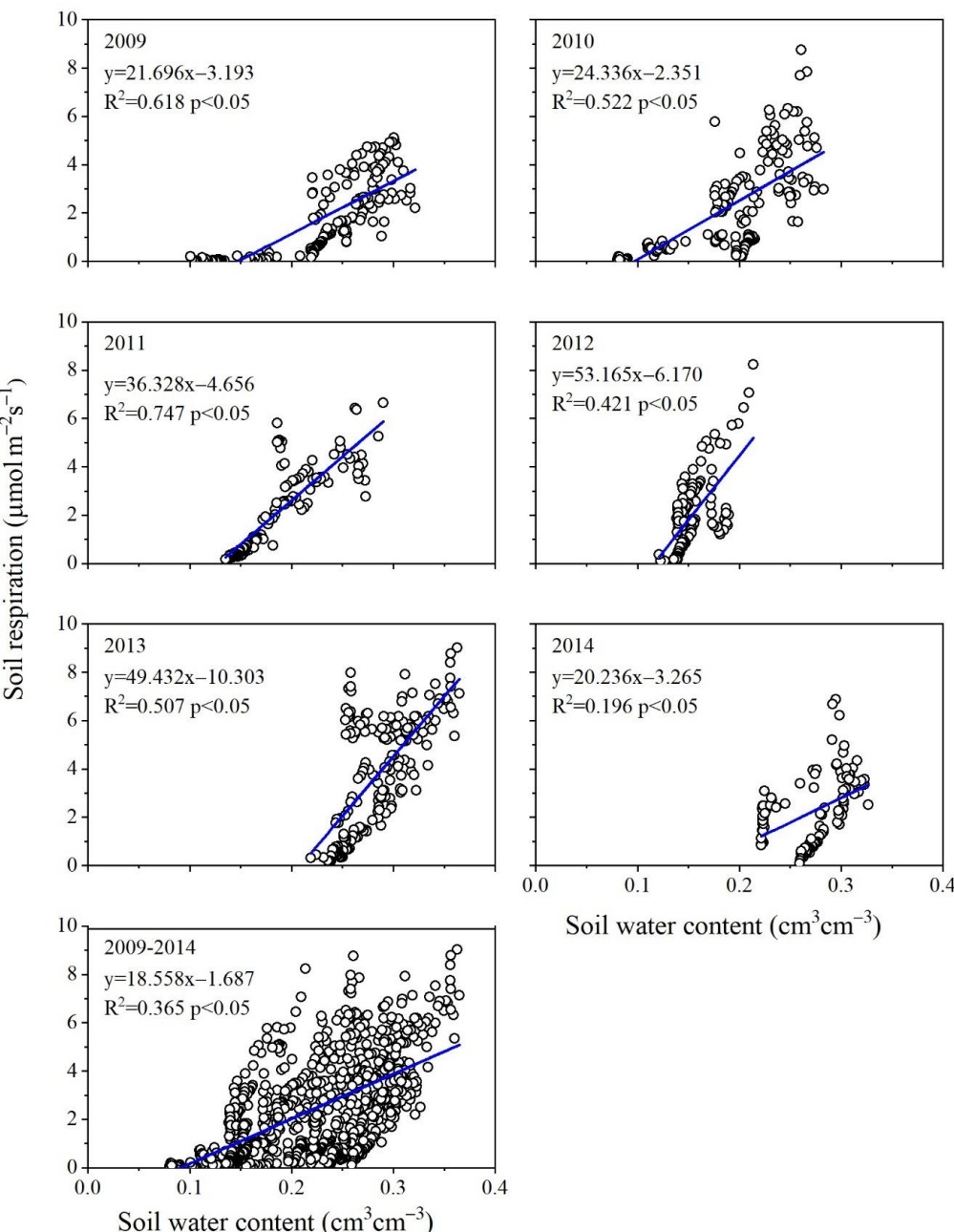

**Figure 3.** Relationships between soil respiration and soil moisture at 10 cm depth during 2009–2014.

### 3.2. Interannual Pattern of Soil Respiration

Across the six years, the annual soil respiration averaged about 526.09 g cm$^{-2}$ yr$^{-1}$ and varied between 391.4 g cm$^{-2}$ yr$^{-1}$ and 597.7 g cm$^{-2}$ yr$^{-1}$ (Figure 4). The interannual variations were relatively slight, with the coefficient of variations (CV) being 13.9%. A regression analysis shows that the annual soil respiration was significantly correlated with soil moisture, whereas it was generally not so with other climatic factors such as soil temperature and rainfall, suggesting that soil moisture was the most predominant factor controlling the annual variations of soil respiration in this meadow steppe ecosystem (Figure 5). However, we failed to detect a significant interannual relationship between the annual total respiration and each of the phytomass components. Nevertheless, the correlation between the annual total respiration and root biomass was apparently more pronounced, displaying a negative trend between each other (Figure 6).

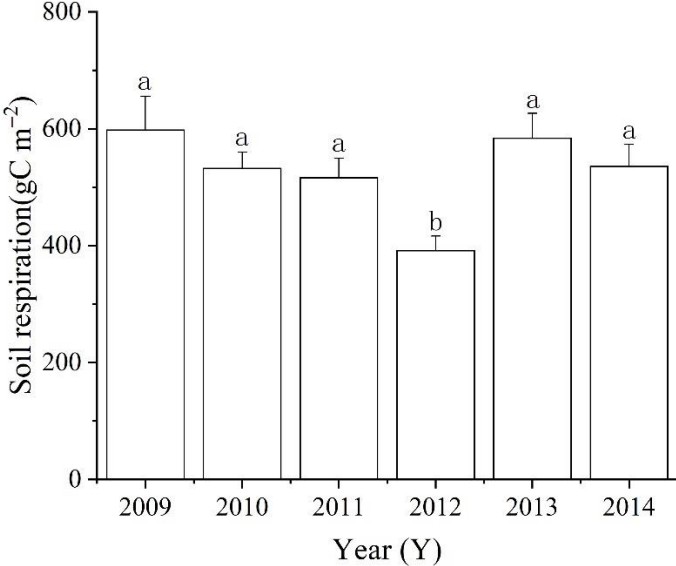

**Figure 4.** Dynamics in annual soil respiration across study years.

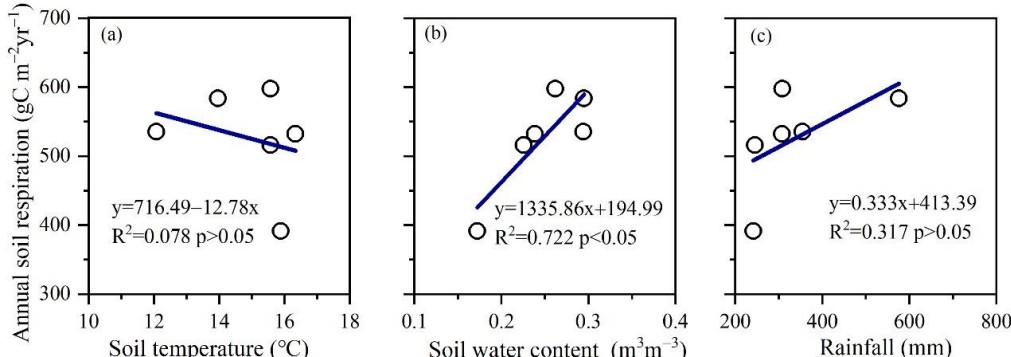

**Figure 5.** Relationships between annual measures of soil respiration and soil temperature (**a**), soil water content (**b**) at 10 cm depth, and rainfall (**c**) during growing seasons.

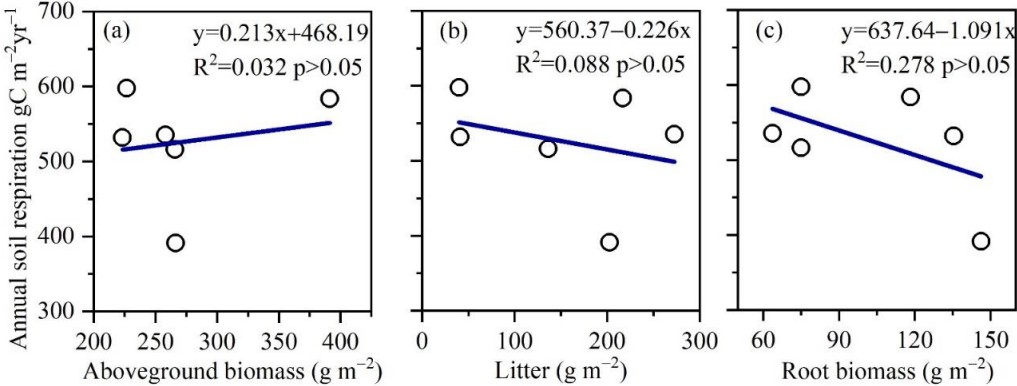

**Figure 6.** Relationships between annual measures of soil respiration and phytomass components: (**a**) above biomass, (**b**) litter, and (**c**) root biomass.

## 4. Discussion

### 4.1. Controls of Seasonal Pattern in Soil Respiration

The seasonal pattern of soil respiration detected in this study is generally similar to what has been reported by most of the analogous studies [11,24,25]. However, the mean value of CVs for within-season soil respiration (0.47) herein was much lower than those

characterizing the typical and desert steppe communities (0.54–0.89) [26,27], while being comparable to those of the alpine meadow steppe (0.54–0.58) [28]. A major reason may consist in that the soil moisture status was much better in terms of both the availability of rainfall and the water-holding capacity of the soil in this study area, which can effectively alleviate the fluctuations in both soil temperature and soil moisture.

Our results show that temperature played a more important role than soil moisture in controlling the seasonal pattern of soil respiration in this steppe. A further analysis exhibited that the proportional contribution of soil temperature ranged from 73% to 95.5%, due to the seasonal variations of soil respiration here. This is presumably attributable to the relatively high rainfall and its even allocation within the growth season of the meadow steppe region, which rendered soil moisture not a limiting factor regulating the seasonal dynamics of soil respiration [29]. Soil water content is a persistent key driver of biological processes for many terrestrial ecosystems, and plays a prominent role in grasslands through its influence on plant productivity [30,31], soil microbial activity [32], and root respiration [33]. Generally, soil moisture would outweigh soil temperature in governing the seasonal dynamics of soil respiration in more droughty regions [34,35], as reported in the desert and xeric steppes [19,36–38] and the typical steppe [11,39,40]. By contrast, at more humid and/or colder steppe regions, just the opposite would hold true, such as that observed in this study area, in the alpine meadow region of the Qinghai-Tibet Plateau [41–43] (and in a humid tallgrass prairie [44].

Furthermore, the seasonal trajectory of soil $CO_2$ flux in temperate grasslands is also influenced to varying extents by plant phenology. In most cases, soil $CO_2$ fluxes often are high early in the growing season when soils are moist; increase to a peak in the mid-growing season (coincident with high temperatures, maximal plant growth, and adequate soil water); and decrease late in the season as a result of decreased plant activity, lower temperatures, and/or depleted soil water reserves [45]. We indeed observed similar patterns in the present study (Figure 1), which may be related to both plant phenology and increased soil water deficits late in the growing season, and a time period featured by the onset of flowering and subsequent senescence of grasses, such that less substrate was available for root respiration. All these suggest that the seasonal pattern of soil $CO_2$ fluxes is likely a result of the interactions among temperature, soil moisture, and vegetation phenology, with their effects on the substrate availability being a basic underlying control on soil $CO_2$ fluxes.

### 4.2. Controls of Interannual Pattern of Soil Respiration

The cumulative soil respiration rate during the growing season (63% of the year from day 100 to day 330) of this study ranged from 559 to 622 g cm$^{-2}$ year$^{-1}$, a range which corresponds well with published estimates for temperate grasslands [8,10,24]. The coefficient of variations in the annual soil respiration rate of this meadow steppe (13.9%) was substantially lower than those observed in semi-arid prairies [14] and alpine meadow steppes [28], and fairly lower than that in a typical steppe [11].

The year-to-year differences in this respect might be a complex reflection of the differences in vegetation type, soil type, and site-specific climatic conditions. A number of studies pointed to the fact that the interannual pattern of soil respiration is more substantially soil moisture- and/or rainfall-dependent [14,16,46], as found in the present study (Figure 5). Parton et al. [47] attested to the fact that large precipitation events promoted carbon uptake, while small precipitation events enhanced heterotrophic respiration, which explains the substantially less-significant relationship between the annual soil respiration and the annual rainfall detected in the present study (Figure 5). Of special note, a negative relationship between the annual soil respiration and the growth season meant soil moisture indeed has been documented in a humid temperate grassland [48]. It should be pointed out that the aforementioned studies mostly lasted for only three years or so, which underscores the need for field observations across more years. In addition, our study showed that soil moisture and temperature together explained a proportion that is much lower than that explained by the soil moisture alone (data not shown), suggesting that counteractive effects

may exist between the two climatic factors in this area. The weak interannual relationship between the annual soil respiration rate and growth season mean temperature observed in the present study may also reflect, to a certain extent, the hysteretic effects of the seasonal soil temperature on the soil respiration across the years [38].

The ambiguous relationship between the peak aboveground biomass and annual soil respiration rate shown in this study is in contrast to those reported by a few other available studies, such as positive [49] and negative [48]. These studies hinted that the peak annual soil respiration rate did not coincide with the maximum aboveground biomass across the study years. Several authors noted that the biomass production of prairie grasslands is primarily a function of timing and quantity of precipitation [16,44]. Furthermore, 2–4 years of antecedent or hysteretic effects of precipitation on ANPP have been widely reported [50–52]. On the other hand, soil respiration appears to be more closely related to the current year or growth season precipitation and/or soil moisture. Heterotrophic soil respiration may be greatly enhanced for one to two days following rainfall events [47]. In the present study, we found that the total soil respiration was most significantly related to the antecedent precipitation amount from October of the previous year to April of the current year across our study years, while aboveground biomass was most significantly related to the precipitation amount from January to July of the current year. In stark contrast, none of the relationships (including root biomass) were significant when the entire year's precipitation was used (Figures 5 and 7). This analysis revealed that an about 2–3 month hysteretic effect of precipitation on the canopy growth existed before its effect on the soil respiration in the study area. Therefore, it is safe to assume that the temporal inconsistency in the response to precipitation between soil respiration and aboveground/root biomass production across the years is likely a mechanism explaining the weak relationship between these parameters.

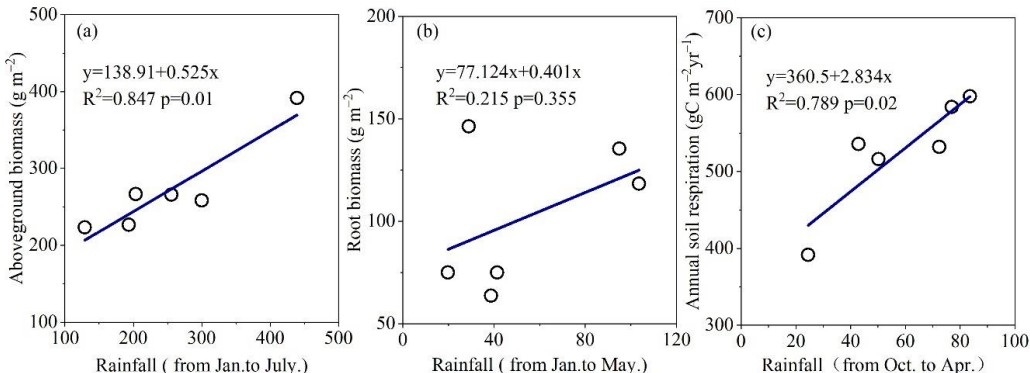

**Figure 7.** Relationships of precipitation measures of different monthly durations with aboveground biomass (**a**), root biomass (**b**), and annual soil respiration (**c**).

## 5. Conclusions

Our findings show that the seasonal pattern of soil respiration was controlled mainly by soil temperature, whereas the interannual pattern was controlled principally by the soil moisture regimes of the meadow steppe ecosystem. These have obvious antecedent effects of precipitation on both the soil respiration and aboveground phytomass in this steppe. Our findings have important implications for better understanding and accurately assessing the carbon cycling of terrestrial ecosystems in response to climate change in a temporal perspective. Future studies should consider the upscale temporal effect of these biophysical drivers on modelling soil carbon exchange in the grasslands.

**Author Contributions:** Conceptualization, X.W. and X.X.; methodology, X.W. and Y.Y.; software, K.F.; validation, X.W., K.F. and B.C.; data curation, B.C. and R.Y.; writing—original draft preparation, X.W.; writing—review and editing, L.L. and Y.Y.; supervision, X.X.; project administration, X.X. and X.W. All authors have read and agreed to the published version of the manuscript.

**Funding:** This study was supported by the National Key Natural Science Foundation of China (32130070), National Natural Science Foundation of China (32171567), and National Key Research and Development Program of China (2021YFD1300502).

**Conflicts of Interest:** The authors declare no conflict of interest.

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
