# Peer review of "Controls of Seasonal and Interannual Variations on Soil Respiration in a Meadow Steppe in Eastern Inner Mongolia"

_agronomy, doi:10.3390/agronomy13010020_

Round 1
Reviewer 1 Report
The manuscript is prepared professionally. It includes a well-crafted abstract and an exhaustive introduction that justifies the research undertaken. The introduction points to the deficiencies in the literature on the subject. The aim is clearly defined. Modern analytical methods were used in the research. Results have important implications for better understanding and accurately assessing the carbon cycling characteristics of terrestrial ecosystems in response to climate change in a temporal perspective. The discussion of the results is well prepared. The conclusions are well-defined. The illustrative material is appropriate.
In my opinion, the manuscript after corrections, will be suitable for publication in a journal.
1- Text writing should be revised, as it seems to be written without Instructions for Authors
2- The abstract should review important objectives, materials, results, conclusions, and applications as concisely as possible. Consider rewriting the abstract
3- Introduction section is need to add some information with current references.
4- A more critical perspective is required on the results obtained such as a more sophisticated account of the implications of the findings and a more advance on the conclusions.
5- The discussion section needs to be tightened and supported with the obtained data and that relevant published studies.
Author Response
Thank you for positive comments and constructive suggestions. We have carefully revised the manuscript and believed it much impoved. Please see the attachment and revised manuscript.

Reviewer 2 Report
- The title is unclear; rewrite it clearly defining the study.
- The abbreviations in the Abstract and all across the draft need to be elaborated the first time they appear.
- The abstract language is very poor. Re-write the abstract with concise information delivery of tangible results (L32-38); Use statistically analysed information/data.
- The introduction section is lengthy and very general. Please summarize it in 3 to 4 medium-size paragraphs. Hypothesis should be clearly defined (L109-117). Use latest references.
- In lines L64-82, “When it comes to the temporal dynamics--------relevant studies” elaborate with the references the expected benefits/rationale in the ‘Introduction’ part.
- Which design was used……is not clear. Please clarify in the summary table of ANOVA. This will allow the readers to understand the type of ANOVA and the significance of observations/treatments.
- Please use only SI units in the results. Please try to limit the number of references to below 50. Cite only the latest-relevant references.
- Table titles need appropriate headings.
- The discussion needs induction of logical reasoning’s with latest references, which is quite lacking in the draft….Improve it.
- PCA analysis between different parameters studied may kindly be worked out if possible to strengthen the findings.
- Conclusion: please be specific and concise the content. Add future prospects at the end. Currently it looks like repetition of results, add quantitative inference too.
Author Response
Thank you for detailed and constructive suggestions. We have carefully revised the manuscript and hope it more publishable. Please see the attachment.

Reviewer 3 Report
The manuscript reports the seasonal and interannual variations of soil respiration in a meadow steppe in eastern Inner Mongolia.
However, some shortcomings were found.
The language needs improvement.
Soil physical properties were not shown. The content of organic matter in the tested soil was not given.
The issue of the activity of rhizosphere organisms was also omitted.
There is no specific information about the vegetation at the research points, about the degree of soil coverage with plants.
Line: 152 “Vegetation traits and soil properties were measured by the standard protocols”. What features were measured and with what standard protocols?
Line 191-199: The text is not comprehensively written, especially the description of table 1.
The methodology does not clearly indicate the control referred to by the authors in the figures (7, 8).
Fig 8. Two separate relationships are shown in Figure 8. The designation and signature should be changed.
Too long and voluminous discussion based on the literature from 20 years ago. The discussion should be shortened to parts directly related to the research.
Author Response

(The authors gave the same response as above.)

Reviewer 4 Report
Good points to present, definetely highlights the importance of root respiration in correlation with soil temperature. The complex approach missed to mention arbuscular mycorrhizal fungi regulation
(see - Zhang, B., Li, S., Chen, S. et al. Arbuscular mycorrhizal fungi regulate soil respiration and its response to precipitation change in a semiarid steppe. Sci Rep 6, 19990 (2016). https://doi.org/10.1038/srep19990 .
The real value of this article should be to draw attention on complex gas-circulation with strong relation of temperature metronomes.
Author Response
Thanks for the positive comments and valuable suggestion. We also added the discussion about the effect of microbiome on soil respiration. In future we would improve our study about fungi regulation mechanism of soil respiration.

Round 2
Reviewer 3 Report
The authors have improved the manuscript sufficiently.